# Cancer-Associated Fibroblast Heterogeneity in Malignancy with Focus on Oral Squamous Cell Carcinoma

**DOI:** 10.3390/ijms25021300

**Published:** 2024-01-21

**Authors:** Julia Arebro, Che-Min Lee, Kevin L. Bennewith, Cathie Garnis

**Affiliations:** 1Department of Interdisciplinary Oncology, British Columbia Cancer Research Centre, Vancouver, BC V5Z 1L3, Canada; cmlee@bccrc.ca (C.-M.L.); kbennewith@bccrc.ca (K.L.B.); cgarnis@bccrc.ca (C.G.); 2Division of ENT Diseases, Department of Clinical Science, Intervention and Technology, Karolinska Institutet, 14186 Stockholm, Sweden; 3Department of ENT Diseases, Karolinska University Hospital, 14186 Stockholm, Sweden; 4Interdisciplinary Oncology Program, University of British Columbia, Vancouver, BC V5Z 1L3, Canada; 5Pathology and Laboratory Medicine, University of British Columbia, Vancouver, BC V6T 1Z7, Canada; 6Division of Otolaryngology, Department of Surgery, University of British Columbia, Vancouver, BC V5Z 1M9, Canada

**Keywords:** cancer-associated fibroblast, CAF subtype, oral squamous cell carcinoma, tumor microenvironment

## Abstract

Oral squamous cell carcinoma (OSCC) remains an understudied and significant global cancer killer and dismal survival rates have not changed in decades. A better understanding of the molecular basis of OSCC progression and metastasis is needed to develop new approaches for treating this disease. The supportive network surrounding cancer tumor cells known as the tumor microenvironment (TME) has gained increasing interest lately since it performs essential protumorigenic functions. Cancer-associated fibroblasts (CAFs) are one of the main cell types in the TME and are known to play a key role in influencing the biological behavior of tumors. CAFs present a heterogeneity both in phenotype as well as functions, leading to the suggestion of different CAF subtypes in several cancer forms. The task to subtype CAFs in OSCC has, however, just begun, and there is today no united way of subtyping CAFs in this disease. This review aims to define the features of CAFs and to summarize CAF subtype research in malignancy with focus on OSCC including aspects as disease prognosis and therapeutic opportunities.

## 1. Introduction

Head and neck cancer consists of several subtypes, with oral squamous cell carcinoma (OSCC) being the most common. The 5-year survival rate for all OSCC patients is approximately 50%, and this measure decreases to approximately 30% when looking only at patients diagnosed with more advanced stages of the disease [1,2]. Frequent late-stage diagnoses and high rates of recurrence are the key drivers of this poor prognosis. Interventions such as immunotherapy and targeted therapies have enhanced cancer treatment efficacy and improved cancer patient survival in general, building on successes from standard care involving surgery, radiation, and chemotherapy. However, gains in survival have been modest. There remains a clear need to develop new approaches for treating this disease. A better understanding of the molecular basis of OSCC progression and metastasis is needed to provide new treatment/disease management options.

Recent years have seen an expanded research focus on the tumor microenvironment (TME) based on the emerging evidence that it plays a crucial role in tumor progression and metastasis. Cancer-associated fibroblasts (CAFs) represent one of the main cell constituents in the TME, and these are known to contribute to stromal desmoplasia, which is linked to worse outcomes for many tumor types including head and neck cancer [3,4]. Expanded research into CAFs has indicated a key role for these fibroblasts in tumor progression and metastasis, making them attractive factors for targeted therapy and immunotherapy research [5]. However, effective targeting of CAFs requires a robust understanding of the heterogeneity of CAF cell populations, as well as the biological and clinical significance of various CAF subpopulations.

CAF subtypes have been widely studied in pancreatic, hepatic, and breast cancer [6,7,8,9,10,11,12,13], yet no tools currently exist for CAF subtyping analysis in the context of OSCC. Moreover, only a few groups have investigated CAF cell population heterogeneity in OSCC. The prognostic role of CAF infiltration in tumors has recently been established: in oral cancer, the presence of CAFs has been linked to negative clinical outcome by several authors [14,15,16,17]. One multicenter retrospective analysis of 282 OSCC patients reported that the strongest independent risk factor of early OSCC death was a feature of stroma rather than tumor cells [18]. Herein, we will review and define the features of CAF subtypes, summarize CAF subtype research in several cancer forms including OSCC, present new potential therapeutic strategies linked to CAF subtypes, and describe future research challenges for CAF subtype research in the context of OSCC.

## 2. Defining Fibroblasts

Fibroblasts exist in almost all tissue types and are the main cell type in stroma that produces the structural framework of human tissue through creation and maintenance of the extracellular matrix (ECM). Fibroblasts are derived mainly from primitive mesenchymal cells, although alternative origins have been reported (this is described in a subsequent section). There is no single marker used to identify fibroblast cells. In practice, fibroblasts taken from tumors/ tissues are usually first defined through negative selection: they do not express epithelial (EpCAM), endothelial (CD31), leukocyte (CD45), or erythrocyte (CD235a) markers. Next, positive expression of mesenchymal markers such as Vimentin (VIM), alpha-smooth muscle actin (αSMA), and others is used to indicate that a given cell is a fibroblast. In culture, fibroblasts can also be characterized by their spindle-like appearance. 

Fibroblast functions include tissue homeostasis, wound healing response through secretion of extracellular matrix proteins to help heal the area, and/or through secretion of cytokines to recruit immune cells and initiate an inflammatory response. Normal fibroblasts can also serve as progenitors for specialized mesenchymal cell types, such as bone-forming osteoblasts and lipid-filled adipocytes [19]. Further, fibroblasts activated in a TME are called CAFs and these are known to display a vast array of functional abilities affecting tumor progression.

## 3. Fibroblast Activation and CAF Markers

### 3.1. Fibroblast Plasticity

CAFs may originate from normal tissue fibroblasts, as reported by others [20], with lineage tracing studies in genetically engineered mouse models supporting this concept [21]. These lineage tracing mouse models utilize a specific fibroblast gene promoter attached to a reporter molecule (e.g., mCherry or GFP). This reporter is expressed once the cell begins to express the fibroblast gene used to promote the reporter in the mouse (e.g., a Col5a1 promoter). The cell continues to express the reporter once activated, and expression of the marker is passed down to all daughter cells. This approach has allowed tracking of a chosen cell type as it morphs into a CAF in a tumor setting. An example of a lineage tracing model from Garcia et al. showed that mesenchymal cells driven by *Hoxb6* and *Gli1* expression become pancreatic fibroblasts but only *Gli1* positive mesenchymal cells become CAFs in pancreatic ductal adenocarcinoma (PDAC) [21]. 

However, lineage tracing studies have also challenged the belief that most CAFs originate from normal fibroblasts in the tissue—Helms et al. have shown that pancreatic stellate cells (PSCs, the resident fibroblast-like cell of the pancreas) that were lineage traced with *Fabp4* only contributed to a fraction of the CAFs in PDAC, contrary to popular belief that most CAFs in PDAC come from PSCs [22]. This work demonstrated that there is heterogeneity in the origin of cells that become CAFs in PDAC and potentially in other tumor types as well. Other than lineage tracing, single-cell RNA-sequencing (scRNA-seq) in tandem with either flow-cytometry sorting or mass spectrometry have allowed researchers to identify specific CAF subsets by surface markers that have distinct gene signatures denoting different functions. For instance, Hutton et al. showed that CD105^+^ and CD105^−^ CAFs in PDAC were mutually exclusive cells that could not be converted into each other and that CD105^+^ CAFs induced increased tumor progression, while less abundant CD105^−^ CAFs supported favorable anti-tumor immunity [23]. CAF origin has been explored in other cancer types than PDAC, and, in addition to in vivo models, many in vitro studies clearly display how fibroblasts can become CAFs [24,25]. These studies have shown many origins of CAFs, including endothelial cells through endothelial–mesenchymal transition (EndMT) [26], epithelial cells under specific circumstances through epithelial–mesenchymal transition (EMT) [27], bone marrow-derived mesenchymal stem cells (MSCs) [28], hematopoietic stem cells (HSCs) [29], cancer stem cells (CSCs) [30], adipocytes [31], pericytes [32], and stellate cells [33]. The impact of these different suspected progenitors on fibroblast activation into certain CAF subtypes during tumorigenesis is currently a busy avenue of inquiry, and the importance of cell origin heterogeneity in CAF effects remains to be explored.

### 3.2. Fibroblast Activation

Fibroblasts can be activated into CAFs through several mechanisms. Among growth factors, the TGFβ family ligands are well-established as activating signals [34]. Lysophosphatidic acid, a lipid mediator, can also drive expression of the myofibroblast marker αSMA while also increasing the activity of the contractile cytoskeleton [35]. Fibroblast activation through Notch signaling [36,37] has been linked to poor prognosis of ductal breast carcinoma [36,37]. Several inflammatory cytokines drive CAF subtype differentiation, with IL-6 activating STAT signaling [38] and IL-1 acting through NF-κB [39]. Stromatogenesis as a result of connective tissue reaction to cancer has been shown to promote desmoplastic fibroblast differentiation in three-dimensional cultures [40], and YAP-mediated enhancement of matrix stiffening helps to maintain CAF phenotypes in breast cancer [41]. 

Physiological and genomic stress can also drive fibroblast activation info CAFs, with factors like reactive oxygen species [42] and double-stranded DNA breaks [43] serving as examples of this phenomenon. There is increasing evidence that cancer cell-derived extracellular vesicles (EVs) can also drive CAF activation [25,44,45,46], though EV content differs and can include miRNAs, TGFβ, and cells surface proteins, as well as other proteins, mRNAs, and lncRNAs [44,47,48]. It should be noted, however, that many CAF subtypes and functions exist on a spectrum, rather than distinctive subtypes—and that these subtypes can have overlapping functions, a reality that amplifies the complexity inherent in defining CAFs subtypes.

### 3.3. Fibroblast Markers

CAFs lack a single, specific biomarker in various tumor types. This can be a significant challenge in the context of CAF research. Besides widely mentioned mesenchymal biomarkers such as VIM, αSMA, fibroblast activation protein (FAP), podoplanin (PDPN), integrin beta-1 (ITGB1, also known as CD29), FSP-1 (S100A4), and platelet-derived growth factor alfa (PDGFR-α), PDGFR-β [7,25], the role of matrix metalloproteinases (MMPs), through which CAFs can facilitate tumor growth, invasion, and metastasis has also been evaluated [49]. It is now commonplace for researchers to try to identify useful CAF markers through either flow sorting followed by RNA-seq and/or scRNA-seq to discover new potential CAF markers that are then validated *in vivo*. The LRRC15 protein represents one example of this—and may also serve as a putative target for precision medicine. High expression of LRRC15 has been reported in CAFs, but not in normal fibroblasts, for multiple tumor types [50]. Moreover, LRRC15 expression has been validated as an important CAF marker that can suggest poor response to immunotherapy treatment [51].

To date, exclusive CAF markers have not been reported. Heterogenous expression of some or all of the above-referenced markers in the context of CAFs is suggestive that tumors arise through distinct CAF subgroups that have different cells of origin, different stages of activation, and potentially unique cell function.

## 4. CAF Subtypes in Various Cancer Types

Flow cytometry and scRNA-seq studies have begun to define the phenotypic and functional heterogeneity of CAFs for individual cancer types, revealing a variety of CAF subtypes (which may also be referred to as “subsets” or “subpopulations”) (Table 1). A cell subtype is defined by specific markers, unique functional properties, and has a secretome pattern that differs from other subtypes. Turley et al. have created a resource for searching CAF and fibroblast markers called “FibroXplorer” [52]. It includes data for both human and mouse CAFs/fibroblasts. This is a scRNA-seq object that was created from pooling data from 17 tissues, 50 datasets, and 11 tumor types. Although this may not help with subtyping CAFs in a specific model or a patient sample, it is a great resource for looking at specific CAF markers in the context of various disease states or tissues of origin. Work by the group behind the FibroXplorer resource has also been used to define two universal fibroblast transcriptional sites of origin using mouse tissues, characterize *Pi16+* and *Col15a1+* expressing fibroblasts, and describe use of dermatopontin (*Dpt*) to trace fibroblasts [53]. While almost all healthy tissue fibroblasts expressed *Dpt*, especially in *Pi16+* and *Col15a1+* expressing fibroblasts, *Dpt* expression has been reported to be lost as a fibroblast becomes specialized either into a CAF or a specific fibroblast niche. While this work is a powerful resource for fibroblast research, emerging findings around CAF subtypes for specific cancer tissues show that there is additional complexity to unpack. Subsequent sections will describe CAF subtype-specific findings for breast cancer, pancreatic cancer, hepatocellular carcinoma, and cholangiocarcinoma with a more comprehensive list along with CAF subtype research in OSCC found in Table 1.

### 4.1. Breast Cancer

Costa et al., Givel et al., and Bonneau et al. have demonstrated the presence of four CAF subtypes in breast cancer when samples were characterized using multicolor flow cytometry [6,7,57]. The CAF-S1 subset, typically positive for FAP and CD29 and lacking CAV1, contributes to immunosuppression through attracting CD4^+^ T cells, increasing CD4^+^CD25^+^ T lymphocyte survival, and promoting their differentiation into FOXP3^+^ cells [7]. This CAF-S1 subset was preferentially detected in aggressive breast cancer [7], in cases of relapsing luminal breast cancer [6], and in mesenchymal high-grade serous ovarian cancer associated with poor prognosis [57]. On the other hand, the presence of the FAP low CAF-S4 subset in metastatic axillary lymph nodes of breast cancer correlated with the later development of distant metastasis, a finding that suggests that evaluating CAF subtypes may be useful as a prognostic marker for recurrence or metastasis. Compared to the CAF-S1 subset, the CAF-S4 subset expressed lower levels of PDGFRβ but the highest level of CD29. CAF-S2 and CAF-S3 were found not only in tumors but also in healthy tissue, suggesting that they are normal-like resident fibroblasts [6].

Using scRNA-seq, Sebastian et al. detected myofibroblast-like CAFs (myCAFs), inflammatory CAFs (iCAFs), and antigen-presenting CAFs (apCAFs) in breast cancer [54]. Interestingly, these subsets with similar transcriptomes could also be detected in pancreatic cancer, based on analyses of publicity available RNA-seq data [9]. The myCAF and iCAF subtypes were also identified by Wu et al., showing strong enrichment of iCAF signature genes to be associated with cytotoxic T lymphocyte (also known as CD4^+^ T cells) dysfunction [55]. In line with this, patients with a low iCAF dysfunction signature level had a significant survival benefit associated with high cytotoxic T lymphocyte levels [55]. MyCAFs, on the other hand, had elevated capabilities for collagen deposition, which has been connected to cancer invasion and disease progress [55]. Using immunohistochemistry (IHC), myCAFs were found to be in close proximity to the invasive tumor interface while iCAFs were in an area distal to this interface but with high presence of tumor infiltrating lymphocytes [55]. Regarding potential clinical significance for the apCAF subtype in breast cancer, an immune modulatory role has been suggested [9,55].

### 4.2. Pancreatic and Gastric Cancer

Pancreatic ductal adenocarcinoma (PDAC) has been the primary subtype of focus of CAF research in pancreatic malignancy, with several authors having delineated CAFs into myCAFs and iCAFs [8,9,10,58]. These two subpopulations of CAFs have been identified using immunofluorescence (IF), IHC, and microscopy [10]. Using scRNA-seq, 962 fibroblast cells could be analyzed in vivo and two distinct subclusters could be formed with unique gene signatures [9]. The subcluster 1 had enriched expression of *Il-6* and *Il-8* and chemokines such as *Cxcl1*, *Cxcl2*, *Ccl2*, and *Cxcl12*, and was identified as iCAFs. The IL-6 expressing iCAFs were located far away from neoplastic cells in the desmoplastic stroma according to IHC [10]. The αSMA positive subcluster 2 was identified as myCAFs and spatially distributed in direct proximity to neoplastic cells forming a periglandular ring surrounding the clusters of cancer cells [9,10]. Using Gene Set Enrichment Analysis (GSEA), enriched pathways could be identified enclosing an upregulation of inflammatory pathways such as IFNγ response, TNF/NF-κB, IL2/STAT5, IL6/JAK/STAT3, and the complement pathway in iCAFs, while myCAFs displayed an upregulation in pathways such as smooth muscle contraction, focal adhesion, ECM organization and collagen formation. Biffi et al. further revealed IL1 to be critical for generation of iCAFs [8]. When analyzing PDAC tumors in Kras^+/LSL-G12D^; Trp53^+/LSL-R172H^; Pdx1-Cre (KPC) mice, a new subpopulation—namely, apCAF—were described [9]. These MHC II-expressing and *Cd74*-expressing apCAFs could activate CD4^+^ T cells in an antigen-specific fashion based on use of an ovalbumin-specific TCR transgenic OTII mouse model. An apCAF subtype has also been suggested by Zhang et al. in HNSCC [62]. 

Another group also divided PDAC CAFs into myCAFs and iCAFs, with IL1 inducing an iCAF phenotype similar to the one reported by Biffi et al. [59]. However, this other group also revealed that hypoxia promoted the iCAF-like state. The TME is characterized by varied extent of hypoxia and associated with poor prognosis of patients with PDAC. This finding links the direct functional impact of hypoxia on CAFs in the TME. A third group also delineated CAFs into myCAFs and iCAFs [60]. They showed in an orthotopic model of diffuse-type gastric cancer (DGC) that EVs from highly metastatic DGC cells transferred various miRNAs and induced chemokine expression (CXCL1 and CXCL8) in fibroblasts.

While most CAF subtypes are studied in their cancer-promoting functions, the discovery of cancer-restraining CAFs (rCAFs) in PDAC, which are defined by the expression of Meflin, complicates our understanding of the role of fibroblasts in cancer. These rCAFs have been reported to have inhibitory effects on tumor progression in pancreas cancer [69]. The rCAFs and the previously mentioned CD105^−^ CAFs in PDAC both show that there are CAFs that help in the fight against cancer and must not be forgotten when identifying and studying heterogeneous CAF subtypes.

### 4.3. Hepatocellular Carcinoma and Cholangiocarcinoma

Ying et al. reviewed studies summarizing CAF subpopulations and their markers in hepatocellular carcinoma (HCC) and cholangiocarcinoma (CCA), highlighting the problematic lack of specific CAF markers [11,12]. Six distinct fibroblast subsets were detected using scRNA-seq on human intrahepatic cholangiocarcinoma (ICC) by Zhang et al.: vascular CAFs (vCAF), matrix CAFs (mCAF), inflammatory CAFs (iCAF), antigen-presenting CAFs (apCAF), EMT-like CAFs (eCAF), and lipofibroblasts. The authors determined that the vCAF subgroup was the most prevalent in the tumors. These vCAFs were shown to secrete IL-6, which induced significant epigenetic alterations in tumor cells and furthered malignancy [13].

## 5. CAF Subtypes in OSCC

Given the emerging complexity and varied reports about CAF subtypes, it is clear that resolving the role of these cells in cancer processes will remain challenging for the near future. It is tempting to suggest that CAF subtypes described by one group are the same as subtypes described by another group with the only difference being the name. For example, Li et al. detected two subtypes in colorectal cancer (CRC) naming them CAF-A and CAF-B [61]. CAF-A expressed markers of myofibroblasts such as *Acta2*, *Tagln*, and *Pdgfa* according to differential expression analysis; however, the authors did not label these as myCAFs. Bonneau et al. stated that their suggested CAF-S1 and S4 myofibroblasts have been validated in other studies, yet no other referred groups use the same nomenclature [6,56]. In fact, they have also shown that CAF-S1 fibroblasts themselves have eight distinct subtypes within that subset, some of which fit the myCAF/iCAF nomenclature [7]. Additionally, another CAF subset that expresses ECM-related genes and displays ECM-remodeling properties has been suggested by several authors but does not have a united name (although the name “matrix CAFs” is repeated by several authors, see Table 2 and Table 3) [13,56,62,67]. One CAF subtype that seems to have an agreed-upon name and function are the inflammatory CAFs (iCAFs) which several authors have described and implicated in OSCC tumorigenesis [6,7,8,9,10,11,12,13,14,15,16,17,18,19,20,21,22,23,24,25,26,27,28,29,30,31,32,33,34,35,36,37,38,39,40,41,42,43,44,45,46,47,48,49,50,51,52,53,54,55,56,57,58,59,60,61,62,63,64,67,69,70]. A recent study demonstrated that OSCC-derived EVs are capable of activating this iCAF subset [25]. Further, it must be noted that CAF subtypes in distribution and functional roles seem to differ between cancer types. Accordingly, it is of high interest to find similarities among different cancer forms, as summarized in Table 2, as well as similarities in CAF subtype transcriptome and pathways (Table 3). In this review, CAFs expressing ECM-related genes have been grouped together although the nomenclature varies.

While several research groups have detected and categorized CAF subtypes in breast, ovarian, pancreatic, and hepatocellular cancer, the work to describe CAF subtypes in OSCC has just begun (Table 1, Table 2 and Table 3). Figure 1 illustrates CAF subtypes identified in several cancer types including OSCC. Galbo et al. attempted to find common ground between CAF subgroupings by analyzing results from head and neck squamous cell carcinoma (HNSCC, most frequently presented as OSCC), melanoma, and lung cancer (LC) [67]. The authors identified six CAF subtypes named pan-CAFs that were shared across the different tumor types. Pan-myCAFs showed elevated expression of activated fibroblast markers and smooth muscle cell markers. Pan-dCAFs revealed elevated expression of genes coding for collagen and extracellular matrix remodeling. Pan-iCAFs and pan-iCAF-2 both showed high expression of genes related to inflammation. The authors suggests that patients with high pan-iCAFs might be linked to a poor clinical outcome due to an enrichment of gene sets and pathways related to interleukin signaling, which is linked to the activation of innate immune cells and the establishment of an immunosuppressive microenvironment [53]. Pan-nCAFs was referred as normal fibroblasts whereas pan-pCAFs showed elevated expression of genes related to cell cycle. The work by Galbo et al. and the FibroXplorer tool represent robust resources for initiating CAF subtype analysis in OSCC models and patient samples [52]. Additionally, Whang et al. [64] used scRNA-seq data dividing CAFs in OSCC into iCAFs and myo-CAFs (mCAFs), with mCAFs expressing αSMA and enriched in vascular smooth muscle contraction resembling the myCAF subtype described by several others [8,9,10,54,58,59,60,67]. The OSCC mCAFs were significantly increased in lymph node metastases compared to the iCAF subtype [64]. The iCAFs were found to be the main source of CXCL12 and correlated to the infiltration of M2 macrophages. The authors thus suggest iCAFs to be central in immune infiltration in OSCC.

Costea et al. detected two CAF subtypes in OSCC. CAF-N having transcriptome and secretome close to normal fibroblasts and CAF-D with more divergent expression pattern [65]. Both subtypes showed tumor-promoting capability but were mechanistically different. CAF-N, secreting significantly higher levels of keratinocyte growth factor (KGF) and acidic fibroblast growth factor (aFGF), defined more of migratory fibroblasts dependent on hyaluronan and essential for supporting cancer cell invasion. CAF-D, synthesizing higher levels of TGF-β1, was shown to increase invasion and expression of EMT markers. This was impaired by TGF-β1 inhibition which suggests CAF-Ds to have similarities with myCAFs. The protumorigenic potential of CAF-derived TGF-β1 was confirmed by Yang et al., subtyping CAFs into myofibroblasts (mCAFs) and inflammatory CAFs (iCAFs) [66]. The authors suggest the two subtypes to both promote OSCC progression but through different regulatory roles in the TME. Patel et al. on the other hand, divided CAFs into C1 or C2 CAFs in gingivobuccal oral carcinoma [68]. C1 CAFs, showing lower expression of αSMA, showed increased proliferation of oral carcinoma cells linked to BMP4. This highlights that αSMA expression is not crucial for protumorigenic CAFs. 

The first steps in detecting an apCAF subtype in OSCC have begun through detecting expression of *Cd74* and other MHC-II related genes [63]. Single-cell transcriptome analysis of a post-chemotherapy squamous cell carcinoma from the oral cavity was used to subdivide CAFs and tumor-associated macrophages (TAMs) [63]. Three clusters of CAFs, namely: iCAFs, apCAFs, and myCAFs, were identified and suggested that the MHC II and CD74 expressing apCAF might improve the tumor immune microenvironment after induction chemotherapy. Another study identified eight clusters of CAFs in HNSCC, combining single-cell and bulk-tissue transcriptome profiles [62]. The authors further linked the data to The Cancer Genome Atlas (TCGA) finding three clusters associated with poorer overall survival. These three clusters were cluster 0, myofibroblasts expressing αSMA and enriched in smooth muscle contraction; cluster 3, expressing ECM-related genes and enriched in epithelial-mesenchymal transition (EMT) gene sets; and cluster 4, suggested to be apCAFs enriched in antigen processing and presentation. The authors suggest the apCAFs participate in immunosuppression in the TME and highlights that apCAFs might also exist in HNSCC [62]. 

CAF subtypes linked to angiogenesis have been detected in several cancer types including ICC, breast cancer, melanoma, and LC as well as various subtypes of HNSCC, including OSCC [13,54,67]. Galbo et al. link the detected myCAF population in melanoma, LC, and HNSCC to angiogenesis through genes upregulated accordingly (*Agfl6*, *Angpt2*, *Pdgfa*), and future studies are encouraged to see whether CAF subtypes can be linked to pathological angiogenesis in OSCC [67].

## 6. The Prognostic Role of CAF Subtypes in OSCC

Looking upon CAFs as a heterogeneous population of cells with different marker proteins, transcriptomes, and secretomes, it follows that different CAFs may have different functional properties—and that they may be associated with tumor progression and clinical features including prognosis. It is suggested that CAFs in HNSCC (including OSCC) are mainly comprised of myofibroblasts, playing a central role in tumor progression through *Akt3* expression [71]. *Akt3* depletion resulted in impaired CAF immunosuppressive activity, potentially counteracting tumor progression, and intratumor infiltration of *Akt3* positive CAFs correlated to unfavorable prognosis among HNSCC patients. Yang et al. defined CAFs by t-SNE plots into myofibroblasts (mCAFs) (highly expressing *Acta2*, *Tagln*, *Mmp11*, *Myl9*, *Postn*, *Tpm1*, and *Tpm2*), and iCAFs (highly expressing chemokines) [66]. The authors could see a correlation between mCAF/iCAF ratio and tumor stage in OSCC, with a higher proportion of mCAFs detected in more advanced stages of disease, suggesting that this is directly related to OSCC progression. However, in regard to patient survival, high expression of mCAF or iCAF genes predicted poor overall survival in OSCC, while high iCAF-related gene expression alone predicted poor overall survival after five years. Zhang et al. [62] also linked expression of myofibroblasts in tumors with poor survival in OSCC patients, while Yang et al. [66] linked myofibroblasts but especially iCAFs to poor overall survival in OSCC. Galbo et al. identified CAF subtypes in melanoma, HNSCC, and LC and associated key molecular pathways in certain CAF subtypes to cancer progression [67]. The molecular pathways deregulated in the tumors with an abundance of certain pan-CAF subtypes revealed significant enrichment for gene sets associated with immune cells activation (with iCAF and iCAF-2 linked to the highest number of associated gene sets) and ECM and collagen (with myCAF, dCAF and pCAF displaying the highest number of associated gene sets). The authors could further link specific pan-CAFs to clinical outcome in some cancer types—high presence of pan-iCAFs correlated to poor clinical outcome in low grade glioma—but pan-CAFs could not be correlated to prognosis in HNSCC. This shows the importance of subtyping CAFs to better understand patient outcomes. Additionally, HNSCC is known to be a very heterogenous group of tumors, both in morphology, etiology, treatment, and prognosis. Thus, it is imperative to explore CAF subtypes in OSCC specifically in order to glean meaningful insights into the role of CAFs in this disease.

## 7. Therapeutic Opportunities Linked to CAF Subtypes

In PDAC, up to 90% of the tumor volume consists of stroma [59]. In HNSCC, CAFs can account for up to 80% of the tumor mass [72]. The tumor–stroma ratio has been shown to have prognostic importance [73], and evaluating this ratio can also garner insights into clinical disease [74]. Tumor stroma may block delivery of therapy to cancer cells, support cancer growth, and promote therapy resistance [48]. On the other hand, the tumor stroma may also contain potential targets for potential therapy. Recently, the number of preclinical experiments targeting CAFs to restore the anticancer immune response has increased dramatically. Further, strategies for CAF-based immunotherapy have emerged: direct CAF depletion including a novel therapeutic that uses FAP-targeting chimeric antigen receptor (CAR)-T cells [5] resulting in reduced CAF activation [75] and functional suppression of CAF-induced ECM remodeling [76,77,78] (Figure 2). 

Multiple rationales exist for targeting CAFs in anti-cancer treatment, although the heterogeneity of CAF populations—and the reality that CAFs are always solely tumor promoting—complicate this approach. Supporting this, recent studies aiming to target stromal cells in general have not always been successful [79,80,81]. A new therapeutic approach would be to embrace the detection of CAF heterogeneity and to understand the importance of subtypes in terms of treatment opportunities. Costa et al. found that silencing B7H3 and CD73, which are highly expressed in S1 CAFs, significantly reduced their impact on the total numbers of tumor suppressive CD25^+^FOXP3^+^ T cells [7]. The authors accordingly suggests that therapy against B7H3 or CD73 could target CAF-S1 cells and enhance anti-tumor immunity by inhibiting CAF-S1-mediated immunosuppression. Alternatively, Biffi et al. suggest a combinatorial therapy with both a TNFα-neutralizing antibody and a IL1 receptor antagonist for targeting iCAFs in vivo [8]. Converting specific CAF subtypes into a less activated state or shifting tumor-promoting CAF subtypes toward a quiescent or tumor-restraining phenotype—or even neutralizing ligands that activate fibroblasts into specific CAF subtypes such (e.g., IL-1, IL-6, LIF, and/or TGF-β)—may open up completely new druggable pathways [82], which is potentially important in OSCC where iCAFs have been shown to have a potential crucial role [25,63,64,66,67].

In terms of targeting specific markers on CAFs in general, FAP is one of the most widely studied. FAP has been used as a target in preclinical therapeutic strategies with vaccines, antibodies, and CAR-T cells [83,84,85,86]. In fact, a promising new FAP-targeting CAR-T cell has been shown to loosen the desmoplasia that restricts T cell invasion into tumors, which enables other tumor-targeting CAR-T cells to be used to finish the job and shrink the tumor [5]. However, FAP does not appear to distinguish the iCAF subtype in PDAC or OSCC [9,25]. 

Numerous previous studies have linked the interaction between CAFs and tumor cells in tumorigenesis. Additionally, infiltrating immune cells like macrophages, dendritic cells, lymphocytes, and neutrophils are another key component in the TME. Cancer immunotherapy has been one of the biggest medical breakthroughs in recent years. Several reports reveal improved patient survival with those who have increased CD3^+^/CD8^+^ T cell tumor infiltration [7,87,88]. The importance of lymphocytes as a positive prognostic factor in the TME has also been studied in oral cancer [89]. The interplay between tumor cells and immune cells throughout tumor progression involves multiple steps with both stimulating and inhibitory factors. The interaction between CAFs, immune cells, and tumor cells in the TME and their role in tumor progression is an ongoing topic of study. It is, however, known that CAFs regulate and enable the pro-tumor activity of TME immune cells both directly and indirectly via the ECM and secretion of cytokines such as IL-6 [39,78,90,91,92]. 

Most research into iCAFs describes them as protumorigenic and prognostically unfavorable, which aligns with the role of iCAFs and IL-6 secretion. Since cancer immunology has largely been focused on T cells and CD3^+^/CD8^+^ T cell response, it is crucial to determine CAF subtype functions in oral cancer, especially linked to CD8^+^ and CD4^+^ T cells as conducted by Costa et al. in S1 CAFs in breast and ovarian cancer [7,57]. For instance, CXCL12 secreted by FAP^+^ CAFs decreases tumor infiltration of CD8^+^ T cells [83]. This makes CXCL12 or its associated signaling pathway a potential therapeutic target. Similar analyses of CAF subtypes in the context of disease will reveal additional therapeutic targets with the potential to impact patient outcomes.

## 8. Conclusions

Herein we have reviewed how heterogenous CAF subpopulations may be defined and the present understanding of how they are generated, particularly in the context of OSCC. The immediate research challenges in this area include (i) development of a unified classification scheme for CAF subtypes in OSCC, (ii) assessment of the relative frequency of each of these subtypes in the context of clinical OSCC tissues and evaluation of the prognostic significance of CAF subtype distribution in OSCC, (iii) functional evaluation of the iCAF subtype in the context of OSCC models, and (iv) an evaluation of therapeutic interventions against specific CAF subtypes. Ongoing work on these areas has the potential to better illuminate our understanding of malignancy and provide new tools for to improve lives of patients.

## Figures and Tables

**Figure 1 ijms-25-01300-f001:**
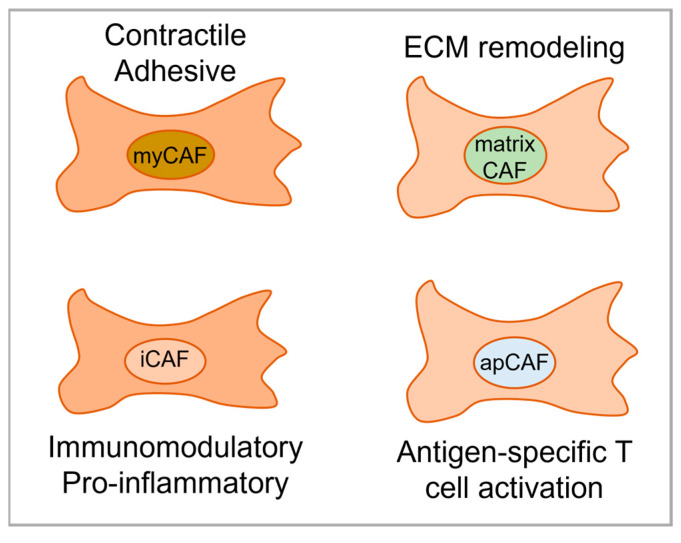
CAF subtypes detected in several cancer forms including OSCC and their possible functions and pathways driving tumor progression. Abbreviations: myofibroblastic CAF (myCAF), inflammatory CAF (iCAF), antigen-presenting CAF (apCAF).

**Figure 2 ijms-25-01300-f002:**
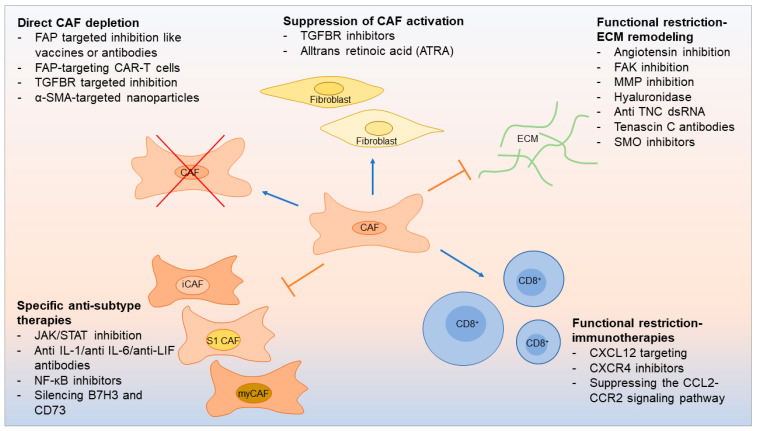
CAF-related therapeutics.

**Table 1 ijms-25-01300-t001:** Differences in CAF subtypes across malignancies.

Cancer Type	CAF Subtype	Subtyping Method	Reference	Year
Breast cancer	CAF-S1, -S2, -S3, -S4	Flow cytometry, IHC *	[6,7]	2018, 2020
	myCAF *, iCAF *, apCAF *. Six clusters	ScRNA-seq *	[54]	2020
	myCAF, iCAF	ScRNA-seq, flow cytometry, IF *	[55]	2020
	vCAF *, matrix CAF, developmental CAF	ScRNA-seq, IHC	[56]	2018
Ovarian cancer	CAF-S1, -S2, -S3, -S4	Flow cytometry, IHC	[57]	2018
Pancreatic ductal adenocarcinoma	myCAF, iCAF	IF, IHC, flow cytometry, RNA in situ hybridization, RNA-seq	[10]	2017
	myCAF, iCAF, apCAF	RNA-seq, RNA in situ hybridization, flow cytometry, IHC	[9]	2019
	myCAF, iCAF	ScRNA-seq	[8]	2019
	myCAF, iCAF	Single-cell digital microdissection	[58]	2019
	myCAF, iCAF	Flow cytometry	[59]	2023
Diffuse-type gastric cancer	myCAF, iCAF	IF	[60]	2019
Intrahepatic cholangiocarcinoma	vCAF, matrixCAF, iCAF, apCAF, eCAF *, lipofibroblast	ScRNA-seq	[13]	2020
Hepatocellular carcinoma	CAF_Port *, CAF_HSC *, CAF_VSMC *	Proteomics, scRNA-seq	[11]	2022
Colorectal cancer	CAF-A, CAF-B (myofibroblast markers)	ScRNA-seq	[61]	2017
Head and neck squamous cell carcinoma	CAF cluster 0-8 (cluster 4: apCAF)	ScRNA-seq	[62]	2021
Oral squamous cell carcinoma	iCAF, apCAF, myCAF	ScRNA-seq	[63]	2022
	iCAF, myo-CAF	ScRNA-seq	[64]	2022
	CAF-N *, CAF-D *	RT PCR *	[65]	2013
	Myofibroblast, iCAF	ScRNA-seq	[66]	2022
Melanoma, head and neck squamous cell carcinoma, lung cancer	myCAF, dCAF, iCAF, iCAF-2, nCAF *, pCAF *	ScRNA-seq	[67]	2021
Gingivobuccal oral carcinoma	C1-type CAF, C2-type CAF	Gene expression microarray	[68]	2018

* Myofibroblastic CAF (myCAF), inflammatory CAF (iCAF), antigen-presenting CAF (apCAF), vascular CAF (vCAF), EMT-like CAF (eCAF), portal fibroblast (Port), hepatic stellate cells (HSC), vascular smooth muscle cell (VSMC), normal fibroblast (CAF-N), divergent CAF (CAF-D), normal fibroblast (nCAF), proliferatory CAF (pCAF), immunohistochemistry (IHC), single-cell RNA-sequencing (scRNA-seq), immunofluorescence (IF), reverse transcription polymerase chain reaction (RT PCR).

**Table 2 ijms-25-01300-t002:** Similarities in CAF subtypes across malignancies.

CAF Subtype	Marker	Additional Notes	Detected in	Reference
myCAF *	MHCII^−^/Ly6C^−^ [9,62]	Activates through: − TGFβ [8]	Pancreatic ductal adenocarcinoma Breast cancer Diffuse-type gastric cancer Oral squamous cell carcinoma Lung cancer Melanoma Head and neck squamous cell carcinoma	[8,9,10,54,55,58,59,60,63,64,66,67]
	αSMA^+^ IL-6^−^ [10]	Function: − Collagen deposition [55]		
	FAP^HIGH^/CD90^HIGH^, αSMA^+^ [55]			
	αSMA [60]			
iCAF *	Ly6C^+^ [9,59]	Activates through: − NF-κB signaling through IL-1 [8] − Cancer-derived EVs [25,60] − Hypoxia induced HIF1α [59]	Pancreatic ductal adenocarcinoma Intrahepatic cholangiocarcinoma Breast cancer Diffuse-type gastric cancer Oral squamous cell carcinoma Lung cancer Melanoma Head and neck squamous cell carcinoma	[8,9,10,13,25,54,58,59,60,63,64,66,67]
	αSMA^−^ IL-6^+^ [10]			
	FAP^LOW^/CD90^LOW^, αSMA^−^ [55]			
	CXCL8 [60]	Function: − Recruits M2 macrophages by secreting CXCL12 [64]		
apCAF *	MHCII^+^ [9,59]	Function: − Activates CD4^+^ T cells [9]	Pancreatic ductal adenocarcinoma Intrahepatic cholangiocarcinoma Breast cancer Head and neck squamous cell carcinoma Oral squamous cell carcinoma	[9,13,54,59,62,63]
Matrix CAF [13] Cluster 3 [62] Matrix CAF [56] dCAF [67]	Fibulin-1 and PDGFRα [56]		Intrahepatic cholangiocarcinoma Head and neck squamous cell carcinoma Breast cancer Melanoma Lung cancer	[13,56,62,67]
S1	CD29^Med^ FAP^Hi^ αSMA^Hi^ FSP1^Low-Hi^ PDGFRβ^Med-Hi^ CAV1^Low^ [6,7,57]	Function: − Promotes immunosuppression [7,57] − Increases migration properties of luminal BC cells [6]	Breast cancer Ovarian cancer	[6,7,57]
S4	CD29^Hi^ FAP^Low^ αSMA^Hi^ FSP1^Hi^ PDGFRβ^Med-Hi^ CAV1^Neg-Low^ [6,7,57]		Breast cancer Ovarian cancer	[6,7,57]

* Myofibroblastic CAF (myCAF), inflammatory CAF (iCAF), antigen-presenting CAF (apCAF).

**Table 3 ijms-25-01300-t003:** CAF subtype-specific transcriptome and signaling pathways.

CAF Subtype	Transcriptome	Pathways
myCAF *	*Acta2*, contractile proteins (*Tagln*, *Myl9*, *Tpm1*, *Tpm2*, *Mmp11*, *Postn*, *Hopx*), *Igfbp3*, *Thy1*, *Col12a1*, *Thbs2* [9]	Smooth muscle contraction, focal adhesion, ECM organization, collagen formation [9]
	*Acta2*, TGFβ response genes (*Ctgf* and *Col1a1*) [10]	Collagen biosynthesis, ECM regulatory pathways [55]
	Contractile proteins (*Acta2*, *Tagln*, *Myl9*, *Tnc*, *Tpm1*, *Tpm2*), growth factors (*Tgfb1*, *Tgfb2*, *Ccn2*/*Ctgf*, *Pgf*, *Vegfa*, *Wnt5a*, *Pdgfrb*), *Igfbp3*, *Cnn2*, *Cnn3*, *Tmem119* [54]	Smooth muscle contraction, vascular wound healing [67]
	*Acta2*, *Fap*, *Pdpn*, *Col1a1*, *Col1a2* [55]	ECM-receptor interaction, vascular smooth muscle contraction, focal adhesion [64]
	Increased *Acta2* and reduced *Cxcl12*, *Des* [58]	Contractile fiber, blood vessel development, tissue migration, ECM organization [66]
	*Acta2*, contractile proteins (*Myh11*, *Mcam*, *Tagln*, *Mylk*) [67]	
	*Pdpn*, *Col1a1* [63]	
	*Acta2*, *Mylk*, *Myl9* [64]	
	*Acta2*, *Tagln*, *Mmp11*, *Myl9*, *Postn*, *Tpm1*, *Tpm2* [66]	
iCAF *	*Il6*, *Il8*, chemokines (*Cxcl1*, *Cxcl2*, *Ccl2*, *Cxcl12*), *Pdgfra*, *Cfd*, *Lmna*, *Dpt*, *Has1*, *Has2*, *Agtr1* [9]	Cytokine signaling, JAK/STAT signaling [8,10]
	Cytokines (*Il6*, *Il11*, *Lif*) and chemokines (*Cxcl1*, *Cxcl2*) [10]	IFNγ response, TNF/NF-κB, IL2/STAT5, IL6/JAK/STAT3, the complement pathway [9]
	*Fbln1*, *Igfi*, *Cxcl1*, *Igfbp6*, *Slpi*, *Saa1*, and complement genes (*C3*, *C7*) and reduced *Acta2* [13]	ECM, inflammatory response regulation, complement activation [13]
	*Ly6c1* and inflammatory cytokines (*Il6*, *Cxcl1*, *Cxcl12*, *Mcp-1*/*Ccl2*, *Mcp-3*/*Ccl7*, *Il33*), compliment pathway members (*C3*, *C4b*, *C1s1*, *C1s2*), *Clec3b*, *Has1*, *Dpt*, *Col14a1*, *Ccl7*, *Ly6a*/*Sca-1*, *Saa3*, *Ogn*, *Prg4*, *Prelp*, *Efemp1*, *Htra3*, *Pdgfra* [54]	Chemotactic regulation, developmental signaling pathways [55]
	*Cxcl12* [55]	Inflammation (iCAF) NFκB signaling pathway (iCAF-2) [67]
	Increased *Vim*, *Fap*, *Col3a1*, *Des*, *Il6*, *Cxcl12* and reduced *Acta2* [58]	TNF signaling pathway, IL-17 signaling pathway, ECM-receptor interaction [64]
	iCAF: *Cfd*, *C3*, *Cxcl14*, *Cxcl12*, *Il33*; iCAF-2: inflammatory genes (*Cxcl2*, *Tnfaip3*), *Icam1*, *Clu*, *Bdkrb1*, *Relb* [67]	Blood vessel development, tissue migration, ECM organization [66]
	*Cxcl12*, *Cxcl14*, *Igf1* [63]	
	*Cxcl12*, *Pdgfra*, *Thy1* [64]	
	Chemokines (*Pdgfra*, *Cxcl12*, *Cfd*, *Dpt*, *Lmna*, *Cxcl2*, *Ccl2*) [66]	
apCAF *	MHC-II genes (*Cd74*, *H2-Aa*, *H2-Ab1*), *Saa3*, *Slpi* [9]	Antigen presentation and processing, fatty-acid metabolism, MYC targets, MTORC1 signaling [9]
	MHC-II (*Cd74*, *Hla-Dra*, *Hla-Drb1*) [13]	Leukocyte cell-cell adhesion, response to IFN-γ, antigen processing, and antigen presentation via MHCII [13]
	MHC-II (*Cd74*, *H2-Aa*, *H2-Ab1*, *H2-Eb*), *Krt8*, *Krt18*, *Fsp1* [54]	Ribosome, antigen processing and presentation, phagosome [62]
	MHC-II (*Cd74*, *Hla-Dra*, *Hla-Drb1*, *Hla-Dqb1*, *Hla-Drb5*, *Hla-Dpa1*, *Hla-Dpb1*) [63]	
Matrix CAF [13] Cluster 3 [58] Matrix CAF [61] dCAF [56]	High levels of ECM signatures (*Col5a1*, *Col5a2*, *Col6a3*), *Postn*, *Fn1*, *Lum*, *Dcn*, *Vcan*) and low levels of *Acta2* [13]	ECM and collagen fibril organization [13]
	*Acta2*^−^ [62]	ECM remodeling (protein processing in the endoplasmic reticulum, oxidative phosphorylation, proteoglycans in cancer, ECM-receptor interaction) [62]
	ECM-related genes such as glycoproteins (*Dcn*, *Lum*, and *Vcan*), structural proteins (*Col14a1*), matricellular proteins (*Fbln1*, *Fbln2*, and *Smoc*), and matrix-modifying enzymes (*Lox* and *Loxl1*), *Cxcl14* [56]	ECM signature [56]
	*Col1a1*, *Col3a1*, *Stc1*, *Mmp1*, *Mmp11*, *Col10a1* [67]	ECM remodeling [67]

* Myofibroblastic CAF (myCAF), inflammatory CAF (iCAF), antigen-presenting CAF (apCAF).

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
