# Peer review of "Cancer-Associated Fibroblast Heterogeneity in Malignancy with Focus on Oral Squamous Cell Carcinoma"

_ijms, 2024, doi:10.3390/ijms25021300_

Round 1

Reviewer 1 Report

Comments and Suggestions for Authors

Due to the poor patient survival of OSCC, there is a critical need to develop new treatment options for this disease. This review discusses the features of CAF subtype and summarizes CAF subtype research in OSCC. More importantly, this article presents new potential therapeutic strategies linked to CAF subtypes and describes future research challenges for CAF-targeting approaches for OSCC. While this review summarizes critical findings in this field, some areas warrant further discussions and edits as shown below.

-       In Section 4.1, “the FAP negative CAF-S4 subset in metastatic axillary lymph nodes …” is more appropriately referred as the FAP low CAF-S4 subset (line 193). Specific marker expressions for CAF-S4 should also be described to better reveal the major distinctions between the S1 and S4 subtypes. Furthermore, the clinical significance of myCAF, iCAFs, and apCaAFs in breast cancer should be included.

-          The significance of temporal evolution as tumors progress and spatial feature of CAF subtypes are inadequately discussed.

Author Response

REVIEWER # 1

Comments and Suggestions for Authors

Due to the poor patient survival of OSCC, there is a critical need to develop new treatment options for this disease. This review discusses the features of CAF subtype and summarizes CAF subtype research in OSCC. More importantly, this article presents new potential therapeutic strategies linked to CAF subtypes and describes future research challenges for CAF-targeting approaches for OSCC. While this review summarizes critical findings in this field, some areas warrant further discussions and edits as shown below.

COMMENT 1: We thank the reviewer for this comment. In Section 4.1, “the FAP negative CAF-S4 subset in metastatic axillary lymph nodes …” is more appropriately referred as the FAP low CAF-S4 subset (line 193). Specific marker expressions for CAF-S4 should also be described to better reveal the major distinctions between the S1 and S4 subtypes. Furthermore, the clinical significance of myCAF, iCAFs, and apCaAFs in breast cancer should be included.

REPLY 1: The text has been edited according to the above suggestion regarding the FAP low CAF-S4 subset (line 201 in the revised manuscript with track changes) and more expression markers have been added regarding the CAF-S4 subset (lines 204-205). A section regarding the clinical significance of myCAFs, iCAFs, and apCAFs in breast cancer has been included (lines 209-219) as well as one lines in each Table 1, 2, and 3 according to reference 56.

COMMENT 2: The significance of temporal evolution as tumors progress and spatial feature of CAF subtypes are inadequately discussed.

REPLY 2: Sections discussing tumor progression (section 6 lines 365-371, 376, 378-387) and spatial features (lines 215-217, 228-232, 313-315) linked to CAF subsets have been added.

Reviewer 2 Report

Comments and Suggestions for Authors

This review manuscript is well-addressed on “heterogeneity of CAFs” of a variety of malignancies, based on a lot of recent publications.

However, authors’ statements regarding “heterogeneity of CAFs in OSCC” are very limited. Although this may be because there have been a few reports on “heterogeneity of CAFs in OSCC”, this reviewer would not think the title is appropriate. In fact, the phrase of “in oral cancer” can be deleted. Or, authors should further concentrate on “CAFs in OSCC”.

This manuscript is too difficult to read because Table 1 does not seem to be consistent with the main text. The size of Table 2 is too large to take a look at, and its structure is very complicated. In addition, there are too many abbreviations in both tables. This reviewer suggests authors should reconsider and rearrange/reconstruct both tables. Or, authors can subdivide the tables for easier-reading.

Author Response

REVIEWER # 2

Comments and Suggestions for Authors

This review manuscript is well-addressed on “heterogeneity of CAFs” of a variety of malignancies, based on a lot of recent publications.

COMMENT 1: However, authors’ statements regarding “heterogeneity of CAFs in OSCC” are very limited. Although this may be because there have been a few reports on “heterogeneity of CAFs in OSCC”, this reviewer would not think the title is appropriate. In fact, the phrase of “in oral cancer” can be deleted. Or, authors should further concentrate on “CAFs in OSCC”.

REPLY 1: The reviewer is correct, the manuscript review on CAFs heterogeneity in malignancy in general with focus on OSCC. The title has been adjusted accordingly to “Cancer-associated fibroblast heterogeneity in malignancy with focus on oral squamous cell carcinoma” (lines 2-3 in the revised manuscript with track changes). Edits have also been made in the Abstract (line 26) and in the main text (line 61, 269, 286-287, 295, 322, 335-336, 362, 462, 465). The sections regarding CAF heterogeneity in OSCC have been expanded as suggested by the reviewer (lines 310-317, 339-358, 365-371, 378-387, 426-427).

COMMENT 2: This manuscript is too difficult to read because Table 1 does not seem to be consistent with the main text. The size of Table 2 is too large to take a look at, and its structure is very complicated. In addition, there are too many abbreviations in both tables. This reviewer suggests authors should reconsider and rearrange/reconstruct both tables. Or, authors can subdivide the tables for easier-reading.

REPLY 2: We apologize for this. Table 1 is a comprehensive list to illustrate differences in CAF subtypes across malignancies, and single works are described more in detail in the main text (clarification added on lines 180-181, 283-284, 295). Abbreviations in Table 1 have now been reduced to make it easier to follow, and the remaining abbreviations are now better described in the legend. Table 2 has been subdivided into two tables (now referred to as Table 2 and Table 3 in the revised manuscript), and the structure of them has been edited slightly to make them easier to follow. Abbreviations have been reduced and a new section in Table 2 states the cancer types each subtype has been detected in, to make the table easier to follow.  

Reviewer 3 Report

Comments and Suggestions for Authors

The authors review the features of CAFs and summarize CAF subtype research in oral cancer including therapeutic options.

The review is excellently written and very well understandable, it is well structured and covers the pertinent and recent literature.

However, specific information on CAFs in oral cancer would merit more impressive presentation, e.g. as additional information in Table 2 and/ or as additional Figure, since CAF heterogenity in oral cancer is the declared topic of the review.

Suggestions:

In Table 2, add information, which CAF subtypes have been observed in which cancer type/ location (the reader will not know only by the reference numbers).

Add one Figure (Table less appropriate) indicating the different CAFs and their functions in oral cancer.

Author Response

REVIEWER # 3

Comments and Suggestions for Authors

The authors review the features of CAFs and summarize CAF subtype research in oral cancer including therapeutic options.

The review is excellently written and very well understandable, it is well structured and covers the pertinent and recent literature.

However, specific information on CAFs in oral cancer would merit more impressive presentation, e.g. as additional information in Table 2 and/ or as additional Figure, since CAF heterogenity in oral cancer is the declared topic of the review.

Suggestions:

COMMENT 1: In Table 2, add information, which CAF subtypes have been observed in which cancer type/ location (the reader will not know only by the reference numbers).

REPLY 1: Table 2 has been edited accordingly and a column regarding which CAF subtype have been observed in which cancer type has now been added.

COMMENT 2: Add one Figure (Table less appropriate) indicating the different CAFs and their functions in oral cancer.

REPLY 2: A figure (now referred to as Figure 1 in the revised manuscript) has been added, displaying CAF subtypes detected in several cancer types including OSCC, and their functions.

Round 2

Reviewer 2 Report

Comments and Suggestions for Authors

Thank you for addressing this reviewer’s previous comments. This reviewer believes that the quality of revised manuscript improved. The manuscript became much easier to follow, especially due to the new table 2 and 3, rearranged from the original table 2.

After the following minor revision, this manuscript should be accepted for publication.

(1) LINE 84 "Origin"; This subheading should be replaced with more appropriate one, or further appropriate words should be added because this paragraph mostly states "pancreas". According to "Origin", authors should state more general issues.

(2) CAF-S2 and CAF-S3 (normal-like resident fibroblasts according to [Ref 6]) appear only in the Table 1, NOT in the main text. Author should explain CAF-S2 and CAF-S3 in the main text as well even though they are not real CAFs.

(3) Please use either "Gingivobuccal oral cancer" in Table 1 or "oral carcinoma" LINE 335, according to [Ref 70].

Author Response

COMMENT (1) LINE 84 "Origin"; This subheading should be replaced with more appropriate one, or further appropriate words should be added because this paragraph mostly states "pancreas". According to "Origin", authors should state more general issues.

REPLY (1) We have replaced the subheading “Origin” with “Fibroblast plasticity” (line 84) and hope the reviewer finds this more appropriate. This section addresses several other cancer types than pancreas, which has been clarified on line 109.

COMMENT (2) CAF-S2 and CAF-S3 (normal-like resident fibroblasts according to [Ref 6]) appear only in the Table 1, NOT in the main text. Author should explain CAF-S2 and CAF-S3 in the main text as well even though they are not real CAFs.

REPLY (2) An explanation on CAF-S2 and CAF-S3 has been added to the main text (lines 205-207).

COMMENT (3) Please use either "Gingivobuccal oral cancer" in Table 1 or "oral carcinoma" LINE 335, according to [Ref 70].

REPLY (3) We have edited this error, using “Gingivobuccal oral carcinoma” both in Table 1 as well as in the main text (line 337).